# Deletion of the *SELENOP* gene leads to CNS atrophy with cerebellar ataxia in dogs

Matthias Christen[1], Sandra Högler[2], Miriam Kleiter[3], Michael Leschnik[3], Corinna Weber[4], Denise Thaller[5], Vidhya Jagannathan[1], Tosso Leeb[1]*

1 Institute of Genetics, Vetsuisse Faculty, University of Bern, Bern, Switzerland, 2 Unit of Laboratory Animal Pathology, Department of Pathobiology, University of Veterinary Medicine Vienna, Vienna, Austria, 3 Department for Companion Animals and Horses, University of Veterinary Medicine Vienna, Vienna, Austria, 4 Laboklin GmbH & Co. KG, Laboratory for Clinical Diagnostics, Bad Kissingen, Germany, 5 Institute of Pathology, Department of Pathobiology, University of Veterinary Medicine Vienna, Vienna, Austria

* tosso.leeb@vetsuisse.unibe.ch

**Data Availability Statement:** All genome sequence data are publicly available. Accessions are given in the S2 Table. All SNV genotyping data are provided with the manucript as S1 File.

## Abstract

We investigated a hereditary cerebellar ataxia in Belgian Shepherd dogs. Affected dogs developed uncoordinated movements and intention tremor at two weeks of age. The severity of clinical signs was highly variable. Histopathology demonstrated atrophy of the CNS, particularly in the cerebellum. Combined linkage and homozygosity mapping in a family with four affected puppies delineated a 52 Mb critical interval. The comparison of whole genome sequence data of one affected dog to 735 control genomes revealed a private homozygous structural variant in the critical interval, Chr4:66,946,539_66,963,863del17,325. This deletion includes the entire protein coding sequence of *SELENOP* and is predicted to result in complete absence of the encoded selenoprotein P required for selenium transport into the CNS. Genotypes at the deletion showed the expected co-segregation with the phenotype in the investigated family. Total selenium levels in the blood of homozygous mutant puppies of the investigated litter were reduced to about 30% of the value of a homozygous wildtype littermate. Genotyping >600 Belgian Shepherd dogs revealed an additional homozygous mutant dog. This dog also suffered from pronounced ataxia, but reached an age of 10 years. *Selenop*-/- knock-out mice were reported to develop ataxia, but their histopathological changes were less severe than in the investigated dogs. Our results demonstrate that deletion of the *SELENOP* gene in dogs cause a defect in selenium transport associated with CNS atrophy and cerebellar ataxia (CACA). The affected dogs represent a valuable spontaneous animal model to gain further insights into the pathophysiological consequences of CNS selenium deficiency.

## Author summary

We studied a form of inherited ataxia in a family of Belgian Shepherd dogs that we termed CNS atrophy and cerebellar ataxia (CACA). Clinical signs were evident at 2 weeks of age and the affected puppies had to be euthanized at 4 weeks of age. The pedigree of the index family with 4 affected and 4 unaffected puppies suggested autosomal recessive inheritance.

**Funding:** The authors received no specific funding for this work.

**Competing interests:** I have read the journal's policy and the authors of this manuscript have the following competing interests: C.W. is employed by a commercial laboratory offering genetic and other diagnostic tests for dogs. The authors declared that no other competing interests exist.

Using a purely positional cloning approach, we identified a complete deletion of the *SELENOP* gene as the most likely causative variant. *SELENOP* encodes selenoprotein P, a protein with multiple selenocysteine residues, which is required for the transport of selenium into the CNS. Selenium measurements in affected dogs demonstrated blood selenium levels of about 30% compared to normal control dogs. Genotyping a cohort of additional Belgian Shepherd dogs with unexplained ataxia identified another CACA case that had a relatively stable clinical condition and reached an age of 10 years. *Selenop*$^{-/-}$ knock-out mice show a related but not identical ataxia phenotype. Our finding of a *SELENOP* gene deletion in CACA affected dogs identifies a spontaneous animal model to gain further insights into the pathophysiological consequences of CNS selenium deficiency.

## Introduction

Ataxias are a heterogeneous group of neurological disorders characterized by irregular and clumsy movements, decreased coordination, tremors, wide-based stance and dysarthria [1]. They are often caused by dysfunction of the cerebellum and then termed cerebellar ataxia [2]. In human medicine, numerous forms of ataxia can be differentiated based on the specific phenotype and mode of inheritance [3–5]. Pathogenic variants causing isolated or syndromic ataxia have been identified in more than fifty genes [4,5].

Dogs share many homologous inherited diseases with humans including different forms of cerebellar ataxia. Veterinary neurology and veterinary diagnostic imaging approaches for dogs made significant advances during the last years enabling comparative investigations that are expected to benefit humans, dogs and other companion animals alike. Currently, less than twenty causative genetic variants for canine forms of cerebellar ataxia are known, but this number is continually growing [6,7].

In Belgian Shepherd dogs, a missense variant in *KCNJ10* encoding a potassium channel causes spongy degeneration with cerebellar ataxia, subtype 1 (SDCA1, OMIA 002089–9615) [8,9]. The clinically similar SDCA2 in Belgian Shepherd dogs is due to a SINE insertion into *ATP1B2* encoding the beta 2 subunit of the Na$^+$/K$^+$ transporting ATPase (OMIA 002110–9615) [10]. Finally, we recently identified a variant in *YARS2* encoding the mitochondrial tyrosyl-tRNA synthetase as candidate causative variant for cardiomyopathy and juvenile mortality (CJM, OMIA 002256–9615) [11]. While CJM is not primarily a neurologic disease, it is characterized by a highly variable clinical phenotype that may also include gait abnormalities. Another form of ataxia in Belgian Shepherds was clinically and histopathologically characterized, but the underlying genetic defect remained unknown [12].

In 2020, a Belgian Shepherd breeder reported a litter with four ataxic puppies. The aim of this study was to characterize the clinical and histopathological phenotype and to identify the underlying causative genetic defect for this presumably new form of ataxia.

## Results

### Clinical description

A Belgian Shepherd litter of the Malinois variety with 8 offspring was investigated. One male and three female puppies presented with ataxia. The other four puppies and the parents were clinically inconspicuous (S1 Fig). Clinical examination was performed on day 27 after birth. The affected puppies showed truncal wobbling, intention tremor, general elevated muscle tone, reduced swallowing reflex, and short episodic spastic fits in variable intensity (Fig 1 and

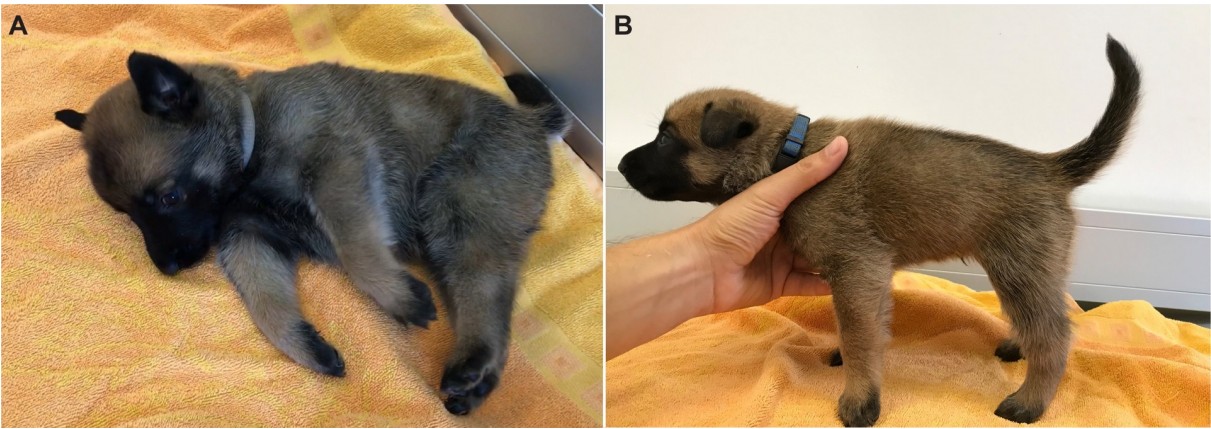

**Fig 1. Clinical phenotype of the affected puppies. A** Female affected puppy in lateral recumbency unable to stand up with severe tremor and episodic fits. **B** Male affected puppy with increased muscle tone of the trunk and neck as well as moderate tremor. More details of the clinical phenotype can be seen in the supplementary data (S1 Video and S2 Video).

S1 and S2 Videos). First obvious signs were observed on post natum day 12 to 14 and developed progressively thereafter. The four affected puppies gained less body weight (mean 1.8 kg) in comparison to the four unaffected littermates (2–3 kg). They were euthanized on day 27 after birth due to animal welfare reasons.

## Necropsy and histopathological examination

During necropsy no gross lesions were detectable except for mild anemia. Histologically, all four animals showed similar lesions in brain and spinal cord to variable extent. In the cerebellum, all cortical layers were atrophic with depletion of Purkinje cells and granule cells (Fig 2A and 2B). Neuroaxonal degeneration was present in midbrain, brain stem and spinal cord. Myelin content was severely diminished in the white matter of brain and spinal cord (Fig 2C and 2D). Gliosis was evident in affected regions showing activation and increased numbers of astrocytes and microglial cells, respectively (Fig 2E–2H). Based on the clinical signs and histopathological changes we propose to designate this phenotype as CNS atrophy with cerebellar ataxia (CACA, OMIA 002367-9615).

## Genetic analysis

The occurrence of ataxia, muscle spasm and fits in multiple puppies from the same litter with healthy parents suggested autosomal recessive inheritance (S1 Fig). We obtained genomic DNA from all ten dogs of this family and performed parametric linkage analysis in the family as well as autozygosity analysis in the four affected puppies. A single ~52 Mb segment on chromosome 4 or roughly 2.2% of the 2.4 Gb dog genome simultaneously showed linkage with a maximum LOD score of 2.31 in the family and shared homozygous genotypes in the four available cases. The exact coordinates of the critical interval were Chr4:28,708,283–80,608,758 (S1 Table).

We sequenced the genome of one of the affected dogs and searched for private homozygous variants by comparing the variants from the case with 735 control genomes (S2 Table). The automated variant calling identified a total of 2.6 million homozygous variants in the genome of the sequenced case, but no private protein-changing variant in the critical interval (S3 Table).

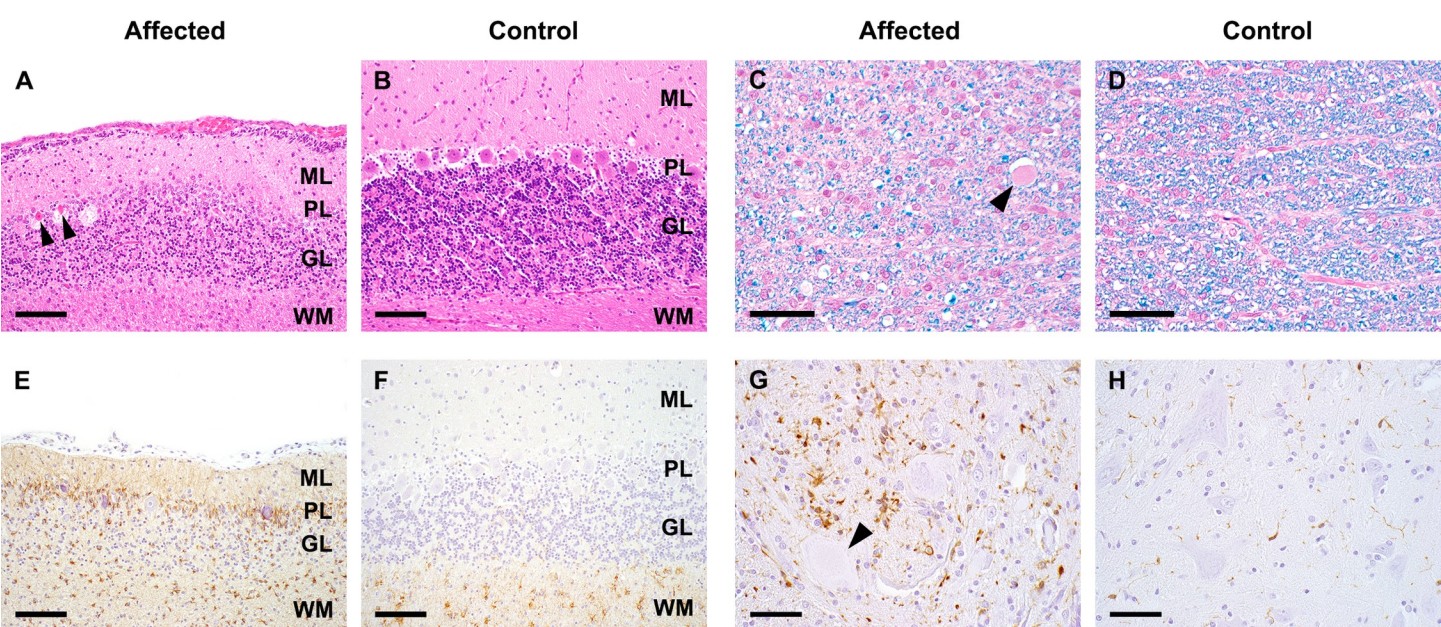

**Fig 2. Histopathological examination. A, B** Cerebellum: Marked attenuation of all cortical layers and necrotic Purkinje cells (arrowheads) (**A**) compared to control (**B**). ML: molecular layer, PL: Purkinje cell layer, GL: granule cell layer, WM: white matter; HE-staining, bar = 100 μm. **C, D** Spinal cord white matter: Diminished myelination (blue staining) and axonal degeneration (arrowhead) (**C**) compared to control (**D**); HE-LFB staining, bar = 50 μm. **E, F** Cerebellum: Increased numbers and activation of astrocytes particularly in Purkinje cell and granule cell layer (**E**) compared to control (**F**); GFAP-IHC, bar = 100 μm. **G, H** Spinal cord gray matter: Increased numbers and activation of microglia associated with neuronal degeneration (arrowhead) (**G**) compared to control (**H**); Iba1-IHC, bar = 50 μm.

Our automated variant calling pipeline considered only single nucleotide variants (SNVs) and small indels. A visual search for structural variants that would have been missed during the initial analysis detected a single structural variant involving protein coding exons in the critical interval. This variant, Chr4:66,946,539_66,963,863del17,325, represents a deletion removing the complete protein coding sequence of the *SELENOP* gene. More specifically, the deletion breakpoints are located ~6.5 kb upstream of the transcription start site of *SELENOP* and within the 3'-UTR of the last exon of *SELENOP* (Fig 3). The deletion was present in homozygous state in the sequenced case and absent from 735 control genomes that were visually inspected in IGV.

We genotyped the deletion in a cohort of 668 Belgian Shepherd dogs. This cohort included the index family with four CACA-affected ataxic dogs, 13 ataxia cases with known pathogenic variants from our earlier SDCA1 and SDCA2 studies [9,10], as well as 20 other unexplained ataxia cases from the Vetsuisse biobank. The genotypes at the deletion co-segregated with the CACA phenotype as expected for a monogenic autosomal recessive mode of inheritance in the index family (S1 Fig).

None of the previously reported SDCA1 and SDCA2 cases carried the *SELENOP* deletion. However, one of the archived unexplained ataxia cases from our biobank was also homozygous for the deletion (Table 1). This dog developed ataxia as a puppy and died at 10 years of age. It reportedly came from a litter with a total of 10 puppies, of which three were euthanized due to severe ataxia at a few weeks of age. The additional case was distantly related to the four affected puppies from the index family (S1 Fig).

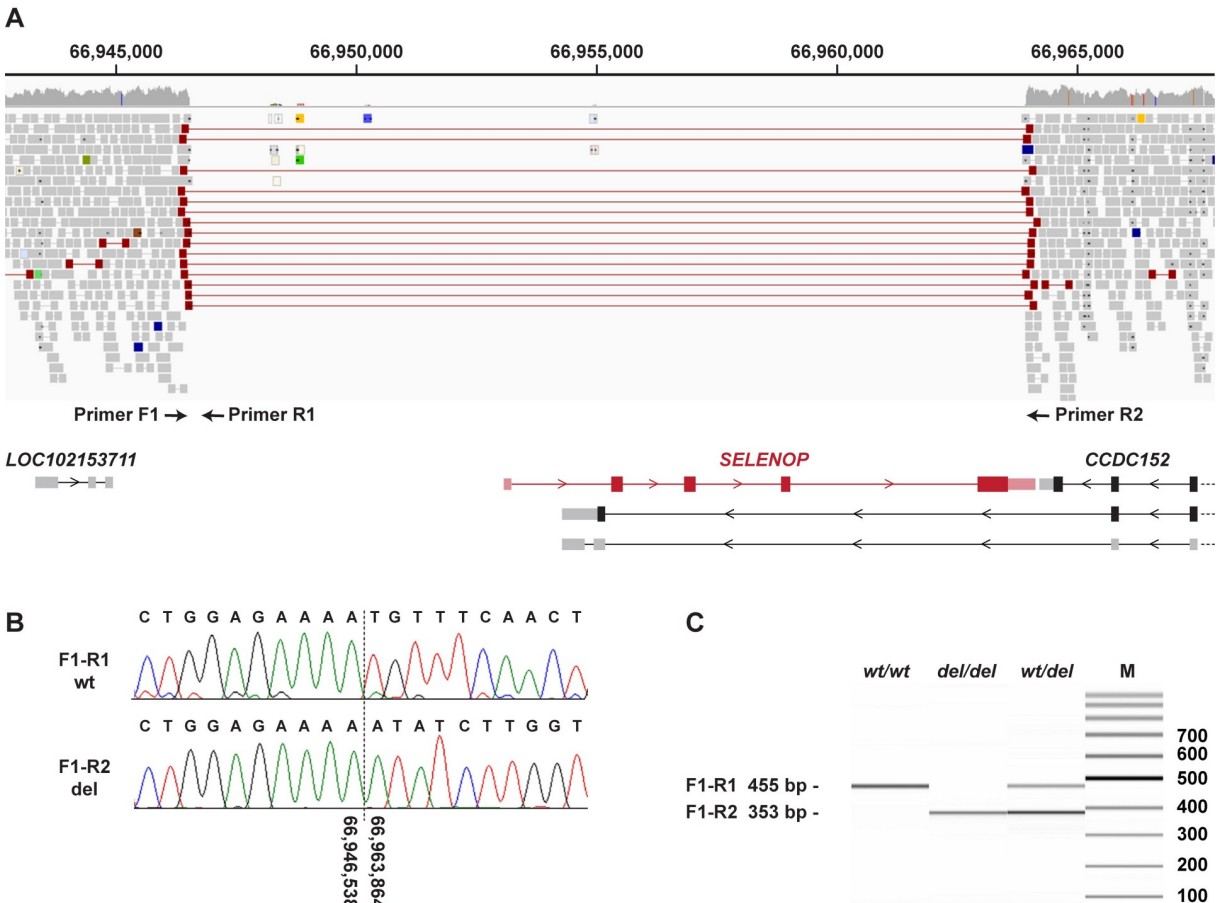

**Fig 3. Details of the Chr4:66,946,539_66,963,863del17,325 variant. A** WGS short-read alignments of an affected dog indicate a homozygous deletion of 17,325 bp. The position of three PCR primers for an allele-specific genotyping assay is indicated. The deletion harbors the entire coding region of *SELENOP*. The neighboring *CCDC152* gene has three annotated transcripts, of which only one is evolutionarily conserved (on top in the figure). The deletion does not extend into this canonical isoform X1 of the *CCDC152* gene (CanFam3.1, NCBI annotation release 105). The deletion affects 3'-exons of two alternative *CCDC152* transcript isoforms whose biological significance is unknown and which are not annotated in humans. **B** Sanger sequencing of the diagnostic PCR products confirmed the deletion breakpoints. **C** Fragment size analysis of the PCR amplification products obtained from genomic DNA of a healthy control (*wt/wt*), an affected dog (*del/del*) and a heterozygous carrier (*wt/del*).

## Selenium measurement

We measured total selenium content from frozen blood samples of the eight puppies. Selenium levels were reduced by ~25% in heterozygous dogs and by ~70% in homozygous dogs ($P_{ANOVA}$ = 0.00011, Fig 4 and S4 Table).

**Table 1. Association of the *SELENOP* deletion with ataxia in 668 Belgian Shepherd dogs.**

| Phenotype | *wt/wt* | *wt/del* | *del/del* |
|---|---|---|---|
| Ataxia cases from the investigated index family (n = 4) | - | - | 4 |
| Ataxia cases affected by SDCA1 or SDCA2 (n = 13) | 13 | - | - |
| Previously unexplained ataxia cases (n = 20) | 19 | - | 1 |
| Controls (n = 631)[a] | 593 | 38 | - |

[a]These dogs do not include any of the 735 control genomes, which were used for the initial analysis.

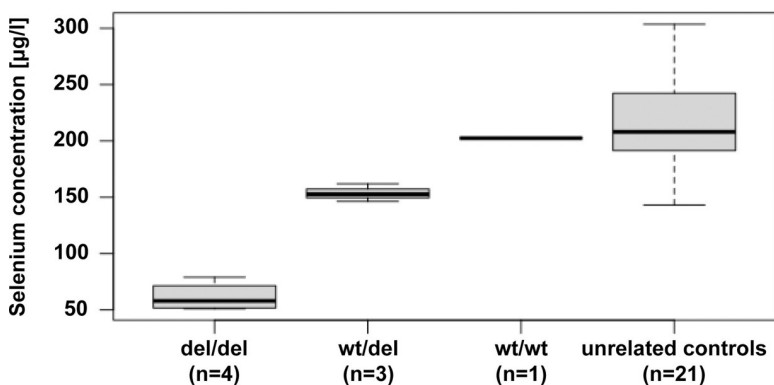

**Fig 4. Selenium concentrations in the blood of dogs with different *SELENOP* genotypes.** The values were derived from the eight puppies of the index family (first three columns) and a cohort of 21 unrelated control dogs (right column).

## Discussion

In this study, we identified a ~17 kb genomic deletion on chromosome 4 harboring almost the entire *SELENOP* gene in Belgian Shepherd dogs with CACA. *SELENOP* encodes selenoprotein P, which functions in storing and transporting selenium [13–18]. Selenoprotein P is the only known protein in vertebrates containing multiple selenocysteine residues [19]. It is primarily synthesized in the liver and secreted into the blood stream [13]. Incorporation of selenium into selenoprotein P prevents the toxic effects of free selenium and the majority of plasma selenium is bound in selenoprotein P [13]. Studies in *Selenop*[-/-] knockout mice demonstrated that selenoprotein P is required for the transport of selenium into the brain and other organs [15–18]. Both blood and brain selenium levels were decreased by approximately 70%-80% in *Selenop*[-/-] mice [15]. Selenoprotein P delivery through the blood-brain barrier is mediated by the low density lipoprotein receptor-related protein 8 (LRP8) [20]. Additionally, astrocytes are able to newly synthesize selenoprotein P [21], which is thought to be transferred to neurons via LRP8. Selenoprotein P can be degraded via the lysosomal pathway to provide selenium for the *de novo* synthesis of other selenoproteins [22]. Selenium is required in several enzymes such as the glutathione peroxidases and thioredoxin reductases that protect cells from oxidative stress [13]. In addition to its established role in selenium transport, it was hypothesized that selenoprotein P might also be directly involved in signaling processes in the brain [23,24].

Similar to the findings in *Selenop*[-/-] mice [15], blood selenium levels in homozygous mutant dogs were reduced to about 30% of the value in a homozygous wildtype littermate. Unfortunately, due to a lack of suitable tissue samples, we could not directly measure the selenium level in brain of the affected puppies. However, previous studies in *Selenop*[-/-] mice demonstrated that their selenium levels in brain were reduced to a similar relative level as those in plasma [15].

*Selenop*[-/-] mice develop ataxia and neurodegeneration when fed low selenium diets [15,16]. Neurological function can stabilize, but not return to normal, when adequate selenium in the diet is reinstituted [17]. *Selenop*[-/-] mice that were fed a diet containing ≥0.25 mg selenium/kg did not develop neurological dysfunction [17]. *Selenop*[-/-] mice show axonal and neuronal degeneration in thalamus, mesencephalon, brainstem and cerebellar white matter [23]. Lesions in these regions are present at weaning and progress when animals are fed low selenium diets. Further lesions develop in the somatosensory cortex and striatum associated with astrogliosis 12 days post weaning upon low selenium diets [25].

Histological lesions present in mesencephalon, cerebellar white matter and brain stem in the *SELENOP*[-/-] dogs were similar to the lesions in *Selenop*[-/-] mice [25,26]. However, in contrast to the phenotype described in *Selenop*[-/-] mice, dogs showed cerebellar atrophy with attenuation of all cerebellar layers and widespread loss of Purkinje cells and granule cells. Cerebellar lesions have been reported in mice with a deletion of the *Trsp* gene encoding selenocysteinyl tRNA [27]. *Trsp*[-/-] mice have a completely abrogated neuronal selenoprotein biosynthesis and show cerebellar hypoplasia/atrophy with Purkinje cell death and decreased granule cell proliferation similar to the homozygous *SELENOP*[-/-] dogs. Similar cerebellar lesions and a wide spread loss of myelin have also been reported in human patients with progressive encephalopathy caused by hypomorphic variants in the *SEPSECS* gene encoding the O-phosphoseryl-tRNA:selenocysteinyl-tRNA synthase, the key enzyme in the sole biosynthetic route to selenocysteine [28].

To the best of our knowledge, there are no reports of human patients with neurological dysfunction caused by genetic variants in *SELENOP* or other spontaneous *SELENOP* mutants in any mammalian species. The deletion in the investigated dogs removed the entire coding sequence of *SELENOP* and may thus assumed to be a true null allele. Consequently, the dogs investigated in this study provide a valuable spontaneous animal model to further study the effects of a complete deficiency of selenoprotein P.

While this study demonstrates the value of clinical veterinary genetics to identify potential spontaneous domestic animal models, we also have to acknowledge some limitations of our study. The studied dogs were privately owned and represented clinical cases. Therefore, the initial sample collection was necessarily limited and precluded e.g. a more detailed characterization of selenium levels in different tissues. A more detailed characterization of the phenotype and possibly modifying genes under controlled dietary selenium intake might further enhance the value of this animal model in the future. However, this will require targeted matings of carrier dogs or the generation of genome-edited *SELENOP* deficient dogs in an experimental setting.

The clinical phenotype in *SELENOP*[-/-] dogs was highly variable in severity. Two out of four affected puppies of the index family showed short fits with tonic muscle spasm of the trunk, neck, and limbs. Intention tremor was observed in all four puppies. Ability to walk was variable, too. One dog remained in lateral or sternal recumbency and any attempt to walk resulted in immediate falling. The remaining three puppies were able to walk with moderate to severe ataxia as well as activity and stress depended extensor muscle spasms. The retrospectively identified fifth case reached an age of 10 years and apparently showed a milder and more or less stable clinical condition. It seems possible that the amount of dietary selenium intake has an influence on the clinical variability. In light of the findings in *Selenop*[-/-] mice, feeding a diet with high and constant selenium levels might be beneficial to affected puppies, if their disease is diagnosed early enough.

## Conclusions

This study identified a deletion of *SELENOP* in Belgian Shepherd dogs with autosomal recessive CNS atrophy and cerebellar ataxia (CACA). Our findings enable genetic testing, which can be used to avoid the unintentional breeding of further affected puppies. The studied dogs might serve as a translational spontaneous animal model to better understand the pathophysiological consequences of *SELENOP* deficiency.

## Materials and methods

### Ethics statement

All examinations were performed after obtaining written informed owner´s consent according to ethical guidelines of the University of Veterinary Medicine Vienna. Blood samples were

collected with the approval of the Cantonal Committee for Animal Experiments (Canton of Bern; permit BE 71/19). All animal experiments were done in accordance with local laws and regulations.

## Necropsy, histology, immunohistochemistry

A full necropsy was performed on all four puppies and samples of brain, spinal cord, sciatic nerves, striated muscle and visceral organs (heart, lung, thymus, liver, kidney, spleen, gastrointestinal tract, pancreas, adrenals and coeliac ganglion) were fixed in 4% neutral buffered formalin for histologic examination. Organ, muscle and nerve samples as well as coronary sections of brain and cervical, thoracic, and lumbar spinal cord were embedded in paraffin and cut at a thickness of 2 μm. All sections were stained with hematoxylin and eosin (HE) and slides examined by light microscopy under a BX 53 Olympus microscope. Furthermore, brain and spinal cord were stained with a combination of HE and Luxol fast blue (HE-LFB) to determine myelination. Brain and spinal cord lesions were compared to juvenile control dogs without CNS lesions, a female Husky puppy with a weight of 1.9 kg and a female French Bulldog puppy with a weight of 2 kg, respectively.

For the detection of astrocytes and microglia immunohistochemistry (IHC) was performed using primary antibodies against glial fibrillary acidic protein (GFAP, Dako, cat# Z0334, dilution 1:10000) and ionized calcium-binding adapter molecule 1 (Iba1, Wako, cat# 019–19741, dilution 1:1250). IHC was performed automatically in an autostainer (Lab Vision AS 360, Thermo Scientific, Freemont, USA) using a secondary antibody formulation conjugated to an enzyme-labelled polymer (Bright Vision Goat anti Rabbit HRP, ImmunoLogic, cat# DPVR 110 HRP). Di-amino-benzidine was used as chromogen and sections were counterstained with hematoxylin.

## Animal selection for genetic analysis

This study was conducted with 668 Belgian Shepherd dog samples. In addition to the Belgian Shepherd index family consisting of four affected and four unaffected full siblings as well as their parents, the genetic study included 658 dogs from different European countries that were donated to the Vetsuisse biobank. 20 of those dogs represented ataxia cases with unknown genetic etiology. Another 13 ataxia cases were due to known pathogenic variants for SDCA 1 or SDCA2. For the remaining 625 dogs, we had no reports of specific neurologic diseases. These were designated as population controls.

## DNA extraction

Genomic DNA was extracted from EDTA blood and hair samples according standard methods using the Maxwell RSC Whole Blood DNA and the Maxwell RSC Blood DNA Kits in combination with the Maxwell RSC instrument (Promega, Dübendorf, Switzerland).

## Linkage analysis and homozygosity mapping

Genotype data for the ten members of the index family were obtained with Illumina CanineHD BeadChips by Geneseek/Neogen (S1 File). For all dogs, the call rate was > 95%. Using PLINK v1.9 [29], markers that were non-informative, located on the sex chromosomes or missing in any of the 10 dogs, had Mendel errors or a minor allele frequency < 0.05, were removed. The final pruned dataset contained 95,207 markers. To analyze the data for parametric linkage, an autosomal recessive inheritance model with full penetrance, a disease allele frequency of 0.5 and the Merlin software [30] were applied.

For homozygosity mapping, the genotype data for the four affected dogs were used. Markers that were missing in one of the four cases, markers on the sex chromosomes and markers with Mendel errors in the family were excluded. The --homozyg and --homozyg-group options in PLINK were used to search for extended regions of homozygosity > 1 Mb. The output intervals were matched against the intervals from linkage analysis in Excel spreadsheets to find overlapping regions (S1 Table). All positions correspond to the CanFam3.1 reference genome assembly.

## Whole genome resequencing

An Illumina TruSeq PCR-free library with ~500 bp insert size was prepared from one affected dog (MA509). We collected 169 million 2 x 150 bp paired-end reads on a NovaSeq 6000 instrument (17x coverage). Mapping to the CanFam3.1 reference genome assembly was performed as described [31]. The sequence data were deposited under study accession PRJEB16012 and sample accession SAMEA7198602 at the European Nucleotide Archive.

## Variant calling

Variant calling was performed using GATK HaplotypeCaller [32] in gVCF mode as described [31]. For private variant filtering we used control genome sequences from nine wolves and 726 dogs. These genomes were either publicly available [33] or produced during other previous projects (S2 Table). To predict the functional effects of the called variants, SnpEff [34] software together with the CanFam3.1 reference genome assembly and NCBI annotation release 105 was used.

## Allele specific PCR and genotyping

We designed an allele-specific PCR with 3 primers for the targeted genotyping of the Chr4:66,946,539_66,963,863del17,325 variant. PCR was performed for 30 cycles using the Qiagen Multiplex PCR kit (Qiagen, Hilden, Germany) in a 10 μl reaction containing 10 ng genomic DNA, 5 pmol primer F1 5'-TGG CAA ATT AAG ATC ACC AGA A-3', and 2.5 pmol each of primers R1 5'- TGA TGA ATT TTT CCC TGA GAC A-3' and R2 5'- CCA CAT TTG GTC AAT TAT GCA C-3'. Product sizes were analyzed on a 5200 Fragment Analyzer (Agilent, Basel, Switzerland). The wildtype allele yielded an amplicon of 455 bp (F1-R1), whereas the deletion allele gave rise to a product of 353 bp (F1-R2).

## Sanger sequencing

After treatment with exonuclease I and alkaline phosphatase, PCR amplicons were sequenced on an ABI 3730 DNA Analyzer (Thermo Fisher Scientific, Waltham, MA, USA). Sanger sequences were analyzed using the Sequencher 5.1 software (GeneCodes, Ann Arbor, MI, USA).

## Selenium measurements

Frozen EDTA blood samples were thawed and centrifuged at 2000 x g for 10 min. Selenium concentrations were measured in the supernatant using atomic absorption spectroscopy (AAS) (ZEEnit 650P, Analytic Jena, Jena, Germany). The intra- and interassay coefficients of variation in dog samples were 1.04–4.58% and 4.74–5.12%, respectively. One-way analysis of variance (ANOVA) was used to test for significant differences between dogs with different genotypes. The significance threshold was set to $p = 0.05$.

## Supporting information

**S1 File. SNV microarray genotypes of 10 Belgian Shepherd dogs (ped- and map-file).**
(ZIP)

**S1 Fig. Pedigree of Belgian Shepherd dogs with ataxia.**
(PDF)

**S1 Table. Linkage and homozygosity data.**
(XLSX)

**S2 Table. Whole genome sequence accessions of 727 dogs and 9 wolves.**
(XLSX)

**S3 Table. Private variants in the genome of the sequenced affected puppy.**
(XLSX)

**S4 Table. Results of the selenium measurements.**
(XLSX)

**S1 Video. Affected puppy showing truncal wobbling and intention tremor.**
(MOV)

**S2 Video. Affected puppy unable to stand, showing mild tremor and severely uncoordinated motion.**
(MOV)

## Acknowledgments

We thank all owners who donated samples and information on their dogs. We thank Nathalie Besuchet Schmutz for excellent technical support. The Next Generation Sequencing Platform and the Interfaculty Bioinformatics Unit of the University of Bern are acknowledged for performing the whole genome sequencing experiment and providing high performance computing infrastructure. We thank the Dog Biomedical Variant Database Consortium (Gus Aguirre, Catherine André, Danika Bannasch, Doreen Becker, Brian Davis, Cord Drögemüller, Kari Ekenstedt, Kiterie Faller, Oliver Forman, Steve Friedenberg, Eva Furrow, Urs Giger, Christophe Hitte, Marjo Hytönen, Vidhya Jagannathan, Tosso Leeb, Frode Lingaas, Hannes Lohi, Cathryn Mellersh, Jim Mickelson, Leonardo Murgiano, Anita Oberbauer, Sheila Schmutz, Jeffrey Schoenebeck, Kim Summers, Frank van Steenbeek, Claire Wade) for sharing whole genome sequencing data from control dogs and wolves. We also acknowledge all canine researchers who deposited dog whole genome sequencing data into public databases.

## Author Contributions

**Conceptualization:** Sandra Högler, Miriam Kleiter, Michael Leschnik, Corinna Weber, Tosso Leeb.

**Data curation:** Vidhya Jagannathan.

**Investigation:** Matthias Christen, Sandra Högler, Miriam Kleiter, Michael Leschnik, Corinna Weber, Denise Thaller.

**Methodology:** Vidhya Jagannathan.

**Supervision:** Tosso Leeb.

**Visualization:** Matthias Christen, Sandra Högler, Michael Leschnik, Tosso Leeb.

**Writing – original draft:** Matthias Christen, Sandra Högler, Miriam Kleiter, Michael Leschnik, Corinna Weber, Tosso Leeb.

**Writing – review & editing:** Matthias Christen, Sandra Högler, Miriam Kleiter, Michael Leschnik, Corinna Weber, Denise Thaller, Vidhya Jagannathan, Tosso Leeb.

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
