## [Decision Letter · Decision Letter 0]

23 Jun 2021

Dear Dr Leeb,

Thank you very much for submitting your Research Article entitled 'Deletion of the SELENOP gene leads to CNS atrophy with cerebellar ataxia (CACA) in dogs' to PLOS Genetics.

The manuscript was fully evaluated at the editorial level and by independent peer reviewers who have been globally convinced by your data. They acknowledged the methodological quality of your analysis, the biomedical relevance of this model and the editorial quality of the manuscript. The three reviewers have made suggestions for improving the manuscript, which you can then incorporate into a revised version. You may take the opportunity to add an additional sentence or a short paragraph in your discussion to address the general comment of Reviewer 2. Alternatively, the Reviewer may simply need a short answer to explain that a dietary selenium deficiency was initially ruled out because of the lack of phenotype in healthy littermates? We let you choose the most appropriate mode of response.

Please note that your revisions should address the specific points made by each reviewer.

[LINK]

Yours sincerely,

Laurent Tiret, Ph.D., D.V.M.

Guest Editor

PLOS Genetics

Scott Williams

Section Editor: Natural Variation

PLOS Genetics

Reviewer's Responses to Questions

**Comments to the Authors:**

Reviewer #1: The manuscript 'Deletion of the SELENOP gene leads to CNS atrophy with cerebellar ataxia (CACA) in dogs' describes a hereditary cerebellar ataxia in Belgian Shepherd dogs clinically detectable around two weeks of age. Histopathology show atrophy in the CNS, especially cerebellum. The authors first identified an associated region using linkage analysis and homozygosity mapping on chromosome 4. After whole genome sequencing of one dog they identified a deletion ( ~17,3kb) containing the SELENOP gene associated with selenium transport to the CNS. In general, the paper is sound and I have no major concerns

Minor issues;

1. Fig 1. The figure is not very illustrative. The authors should consider to exchange the figure with videos if possible/available.

Under “Necropsy and histopathological examination”- the authors have not explained what is used as a healthy control (breed/age)? Please add.

2. Fig 3. The figure contains a lot of information but is a suboptimal. The primers F1/R1 and R2 could be moved/lifted closer to the IGV-sequence. The graphical presentation/naming of the three CCDC152 could be improved.

3. Selenium measurements was performed in frozen blood samples of the eight puppies and shows a significant reduction in affected and heterozygotes. It is mentioned that SELENOP “is required for the transport of selenium into the brain and other organs”. How well is blood concentration of selenium in this case a good expression for the tissue content? Does “required” mean that it is expected that the concentration in brain is “close to zero” when the gene is deleted (or do the authors mean “involved in transport of selenium”)? The clinical variation is interesting, but maybe a variation in selenium-supply to puppies would not be big within the same litter?

The authors should shortly discuss the value of blood selenium in the situation where the involved gene has a role of transport into CNS, and why selenium was not measured in brain (lack of appropriate tissue-samples?). Also, if there is any knowledge about correlation between blood and tissue concentration of selenium in knockout mice? Could other genes be responsible for the clinical variation (like the mentioned LRP8 and Trsp) or do the authors believe that selenium supply is the only variable?. If such information is missing the value of spontaneous models might increase.

Reviewer #2: The paper by Christen et al. describes the natural deletion of SELENOP in dogs as a potential new model of studying Se-metabolism. The topic and the study design facilitates the publication of this manuscript with the current journal, in my opinion. The paper uses adequate methodology and is well written regarding language, style and word usage. I would recommend a minor revision, however, a section of the study limitation is currently missing so I have to recommend a major revision at this point. Please, disclose the limitation of your study (and strengths if you wish so too) in a separate section or just as a part of your discussion. One of the limitation is probably unknown Se intake for the progenitor dogs and offspring as an example. That is my only major comment.

Minor comments

1) I would recommend omitting CACA abbreviation from the title, you have it in the abstract for indexing purposes already.

2) Running title is irrelevant, I would say "SELENOP deletion leads to cerebellar ataxia in dogs" highlights the content of the paper much better (still <70 characters).

3) Necropsy and histopathological examination section: State here which tissues were examined in brief, refer to Material section if needed to avoid major repeats. Otherwise, it is hard to follow. In line with that, testes are severely damaged in Selenop-/- mice leading to male infertility even under Se-rich diets (see studies by Schomburg's group at Charité and other groups). Was it investigated in SELENOP deleted dogs? Any plans to go into that later, if not?

4) Discussion:

a) the second sentence: Mention here which chromosome contains SELENOP gene in dogs as a service to the reader.

b) the third sentence: To be accurate, this is true for vertebrates but not generally true.

c) "To the best of our knowledge, there are no reports of human patients with genetic variants in SELENOP or other spontaneous SELENOP mutants in any mammalian species", I think SNPs of SELENOP gene in humans may be worth mentioning in this section. There were a few relevant studies about that (e.g. by Méplan's group). Additionally, Bellinger's group hypothesised that SELENOP might have some signalling functions in the brain through LRP8 (doi: 10.3233/JPD-2012-11052), which may be partially responsible for the neurological phenotype observed.

d) Necropsy, histology, immunohistochemistry section: space missing – 2μm.

Reviewer #3: This paper presents the straightforward identification of a large deletion associated with a central nervous system (CNS) atrophy and cerebellar ataxia in Belgian Shepherd dogs. The identification of the variant is convincing. It was performed using relevant approaches combining linkage, homozygosity mapping and genome sequencing. The identified variant causes the whole deletion of SELENOP, a gene previously associated with ataxia in mice and involved in selenium transport in the CNS. Despite a small number of cases, clinical and histopathological findings as well as biological measurement of selenium level strengthen the involvement of the identified mutation in the phenotype. This is the first description of a SELENOP loss of function (null allele) in a non-rodent animal model.

The paper is well structured and well written. I have only minor comments to make on the paper.

Abstract

Is CNS (central nervous system) a commonly used abbreviation? Otherwise, please define.

Please replace wt/wt by homozygous wildtype.

Introduction

Reference 6 (2014) could benefit to be completed by a reference to the OMIA web site (more recent update).

Results

Linkage analysis does not reach statistical significance threshold. Max LOD score < 3. Please add this information in the text.

The fifth affected dog that is homozygous for the deletion is related to the four affected puppies (Fig S1). Please add this relevant information in the text.

Supplementary Materials

S1Table: Please add some information:

- LOD score values for first and last SNV from the linked interval

- ID of the SNV with the maximal LOD score

- legend for the last four columns of the "Homozygosity visual Chr4" table

**Have all data underlying the figures and results presented in the manuscript been provided?**

Reviewer #1: Yes

Reviewer #2: Yes

Reviewer #3: Yes

PLOS authors have the option to publish the peer review history of their article (what does this mean?). If published, this will include your full peer review and any attached files.

Reviewer #1: No

Reviewer #2: **Yes: **Nikolay Solovyev

Reviewer #3: No

---

## [Editor Report · Decision Letter 1]

12 Jul 2021

Dear Dr Leeb,

Your responses and your revised version have been carefully read and validated. We are now pleased to inform you that your manuscript entitled "Deletion of the SELENOP gene leads to CNS atrophy with cerebellar ataxia in dogs" has been editorially accepted for publication in PLOS Genetics. Congratulations! 

We hope that the revision process will have resulted in a version that will further highlight the quality of your original work.

Yours sincerely,

Laurent Tiret

Guest Editor

PLOS Genetics

Scott Williams

Section Editor: Natural Variation

PLOS Genetics

Comments from the reviewers (if applicable):

**Data Deposition**

http://datadryad.org/submit?journalID=pgenetics&manu=PGENETICS-D-21-00621R1

**Press Queries**

---

## [Editor Report · Acceptance letter]

28 Jul 2021

PGENETICS-D-21-00621R1 

Deletion of the SELENOP gene leads to CNS atrophy with cerebellar ataxia in dogs 

Dear Dr Leeb, 

We are pleased to inform you that your manuscript entitled "Deletion of the SELENOP gene leads to CNS atrophy with cerebellar ataxia in dogs" has been formally accepted for publication in PLOS Genetics! Your manuscript is now with our production department and you will be notified of the publication date in due course.

With kind regards,

Katalin Szabo

PLOS Genetics

On behalf of:
